# Evaluating the Effects of UAS Flight Speed on Lidar Snow Depth Estimation in a Heterogeneous Landscape

**Franklin B. Sullivan** [1,*] , **Adam G. Hunsaker** [1,2] , **Michael W. Palace** [1,3] **and Jennifer M. Jacobs** [1,2]

1. Earth Systems Research Center, Institute for Earth, Oceans, and Space, University of New Hampshire, Durham, NH 03824, USA; adam.hunsaker@unh.edu (A.G.H.); michael.palace@unh.edu (M.W.P.); jennifer.jacobs@unh.edu (J.M.J.)
2. Department of Civil and Environmental Engineering, University of New Hampshire, Durham, NH 03824, USA
3. Department of Earth Sciences, University of New Hampshire, Durham, NH 03824, USA
* Correspondence: franklin.sullivan@unh.edu

**Abstract:** Recently, sensors deployed on unpiloted aerial systems (UAS) have provided snow depth estimates with high spatial resolution over watershed scales. While light detection and ranging (LiDAR) produces precise snow depth estimates for areas without vegetation cover, there has generally been poorer precision in forested areas. At a constant flight speed, the poorest precision within forests is observed beneath tree canopies that retain foliage into or through winter. The precision of lidar-derived elevation products is improved by increasing the sample size of ground returns but doing so reduces the spatial coverage of a mission due to limitations of battery power. We address the influence of flight speed on ground return density for baseline and snow-covered conditions and the subsequent effect on precision of snow depth estimates across a mixed landscape, while evaluating trade-offs between precision and bias. Prior to and following a snow event in December 2020, UAS flights were conducted at four different flight speeds over a region consisting of three contrasting land types: (1) open field, (2) deciduous forest, (3) conifer forest. For all cover types, we observed significant improvements in precision as flight speeds were reduced to 2 m s$^{-1}$, as well as increases in the area over which a 2 cm snow depth precision was achieved. On the other hand, snow depth estimate differences were minimized at baseline flight speeds of 2 m s$^{-1}$ and 4 m s$^{-1}$ and snow-on flight speeds of 6 m s$^{-1}$ over open fields and between 2 and 4 m s$^{-1}$ over forest areas. Here, with consideration to precision and estimate bias within each cover type, we make recommendations for ideal flight speeds based on survey ground conditions and vegetation cover.

**Keywords:** snow depth; lidar; UAS

## 1. Introduction

Globally, approximately 37% of the Earth experiences non-persistent snow with forests compromising 20% of those seasonal snowpacks in the Northern Hemisphere [1,2]. Winter processes in forests, as compared to open areas, are distinguished by interception, lower albedo, distinct energy balances that enhance or diminish ablation and reduced wind-driven snow redistribution with unique processes at the forest edges [3–6]. Among forests, canopy density, forest structure, forest management and climatic region affect those processes with considerable variability within and across sites [7–12]. This variability makes it difficult to obtain meaningful snow depth measurements over local watershed scales. While models have increased their complexity to capture forest snow processes and demonstrated the value of modeling forest snow at a 1 to 2 m spatial resolution [2,13–17], it remains challenging to validate model results and to provide realistic distributions of highly variable snow depth in forests [18]. High-resolution snow depth observations are required by the snow community to develop and validate distributed snow processes

modeling frameworks, improve stream flow predictions, and to provide better estimates of snow water equivalent (SWE) for water resource planning [7,19].

Airborne laser scanning (ALSM) has a history of providing value in mapping large areas at 1 m resolutions [13,20–22], but has issues with observation gaps in forested regions due to low point density, which may limit the characterization of snow processes [8,13,18,23]. Unpiloted aerial vehicle-lidar (UAV-lidar) can provide opportunistic, moderate-cost snow depth observations at scales intermediate to in situ observations and satellite remote sensing, but over smaller areas than ALSM [18]. As compared to ALSM's above canopy point cloud density on the order of 0 to 20 points/m$^2$, UAV-lidar routinely provides a below canopy point cloud density that is 10 to 100 times higher, has fewer forest grid cells with missing data, and yields snow depth observations on cm scales [24–26]. Observations at these scales allow processes such as snow-vegetation interactions, solar radiation, and wind redistribution to be discerned [25,27,28].

UAV-lidar has shown that forest grid cells have greater variability as compared to field cells and require ground return point counts on the order of 50 points/m$^2$ to produce snow depth estimates with 2 cm confidence intervals [26]. It has been demonstrated that greater accuracy is achieved through UAS-lidar as compared to photogrammetry, particularly over dense vegetation and forested regions [29]. This is the result of lidar pulse characteristics (e.g., beam divergence, scan angle) that allow for penetration through small gaps in the canopy to the forest floor, whereas the features present in aerial imagery of forested areas are limited by narrower observation angles, ground sampling distance, and vegetation characteristics. Nevertheless, in most previous studies lidar snow depth maps contain gaps due to missing data. Gap filling to create a continuous map has been recommended [15], but may introduce overestimates of snow depth in dense foliage areas or introduce unrealistic snow depth values [28,30].

Understanding the effects of snow-cover, or its absence, on lidar return retrieval under different vegetation canopies can aid in improving snow depth estimation by better defining optimal acquisition methods. There may be differences in return density under snow-free and snow-cover conditions that contribute to uncertainty in snow depth measurements. For example, it was found that point cloud density may be lower for snow-covered surfaces as compared to bare surfaces [31]. In mixed vegetation landscapes, differences in a vegetated canopy's structure result in a differential in returns among vegetation types. Scattering caused by tree canopies can result in reduced return rates of laser pulses, with ground surface returns predominantly occurring in canopy gaps [32]. Compensating for lost returns in forested environments can be accomplished by flying slower, but this will likely result in excessive returns in open areas. Additionally, it is unclear whether the increased point cloud density from slow flight speeds will oversample canopy gaps rather than add new information.

This study investigates the influence of UAS flight speed on the ability to collect accurate high spatial resolution snow depth data. The purpose of conducting this study was to determine how to best optimize UAS-lidar flight planning to achieve consistent and acceptable levels of uncertainty in snow depth products over different land types. We compare data acquired at four flight speeds for both snow-on and baseline (i.e., snow-off) conditions over deciduous forests, coniferous forests, and open field land types. UAS lidar snow depth estimates are compared to in situ observations collected over multiple 1 m grid cells. The within grid cell distribution of lidar returns are examined to quantify any systematic sampling biases, and finally, we examine how the presence of snow affects observed patterns.

Our guiding hypotheses are that lidar pulse loss rate is related to vegetation cover type and that adjusting the relative flight speed over different cover types can be performed to account for pulse loss rate to produce point clouds with constant or similar return rates. Resulting from the increased reflectance of a snow surface compared to bare earth at the operating wavelength of our laser scanner, we also hypothesize that snow -increases ground return rate and total return rate, with higher relative increases in forest

settings, where leaf litter and low-lying vegetation contributing to lost pulses are covered by snow. A definite combination of speeds in snow-free and snow-cover conditions result in the desired confidence intervals of snow depth estimates over different vegetation cover. The combination of speeds differs across cover types and with consideration to speed/density/confidence trade-offs, we can identify speed combinations in different vegetation types in snow-free and snow-covered conditions that are best for snow depth mapping in different landcover types.

## 2. Materials and Methods

### 2.1. Study Site

This study was conducted at Thompson Farm Research Observatory in Durham, New Hampshire, United States (43.10892°N, 70.94853°W, 35 m above sea level). Repeat UAV flights were completed over a 1.8 ha subsite of Thompson Farm consisting of mixed deciduous/conifer forest and an open agricultural field. The primary conifer species within the forest were white pine (*Pinus strobus*) and Eastern Hemlock (*Tsuga canadensis*), and the most abundant deciduous species present were northern red oak (*Quercus rubra*), red maple (*Acer rubrum*), shagbark hickory (Carya ovata), and white oak (*Quercus alba*). Gentle to moderate elevation gradients are found throughout the study site. Over the past 20 (2003–2023) winter seasons (January–March), the mean, maximum, and minimum daily air temperatures were −2, 3, and −7 °C, respectively. Over the same period, the average daily winter season precipitation was 3.1 mm, with an average of 33 winter season days per year with recorded precipitation.

### 2.2. UAS Lidar Acquisition

A series of UAS lidar surveys were conducted over the study site during December 2020. First, with leaf-off conditions, the snow-free lidar data set was acquired on 11 December 2020 (from here on referred to as the baseline survey). On 18 December 2020, a lidar survey was conducted following precipitation on 17 December that resulted in a snow depth of approximately 11 cm at our study site (from here on referred to as the snow-on survey). The snowpack remained dry between deposition and lidar acquisition as the air temperature was consistently below 0 °C with an average temperature of approximately −5 °C. Survey flights consisted of two parallel 250 m flight lines spaced 40 m apart, flown at an altitude of 65 m. For each survey, both flight lines were flown at 2, 4, 6, and 8 m s$^{-1}$ to yield varying return densities.

Lidar data were acquired over the study area using a VLP-16 sensor (Velodyne Lidar, San Jose, CA, USA), with direct georeferencing performed by time-synchronizing returns with GPS and IMU data collected by an APX-15 UAV INS (Trimble Applanix, Richmond Hill, ON, Canada). Following postprocessing, APX-15 UAV INS data has 2–5 cm positional, 0.025° roll and pitch, and 0.08° true heading uncertainties, with a measurement rate of 200 Hz, allowing for direct georeferencing of lidar returns based on laser pulse timestamps [26]. Postprocessing in POSPac Mobile Mapping Suite (MMS) UAV (Trimble Applanix, Richmond Hill, ON, Canada; v. 8.2.1) resulted in approximately 3 cm positional accuracy for both flights. The VLP-16 sensor is a 16-channel lidar sensor with a 30° vertical field of view and rotating lasers that are spaced evenly between −15 and +15°. Each channel rotates to provide a horizontal field of view of 360°. The VLP-16 collects up to 300,000 points per second with an accuracy of ±3 cm at a range of 100 m. The sensor was mounted with the vertical field of view parallel to the ground. Lidar data were recorded to a data storage system, the HyperCube (Headwall Photonics Inc., Bolton, MA, USA). The payload was mounted in a fixed position on the underside of a DJI Matrice 600 Pro UAV (Shenzhen, China). Point clouds were filtered in R to remove non-ground returns using the Progressive Morphological Filter algorithm [33] as implemented in the lidR package. The resulting point clouds were gridded to develop 1 m spatial resolution digital terrain models (DTMs) and return density maps.

### 2.3. Vegetation Classification

A land classification map was derived from optical red-green-blue (RGB) imagery that was collected over the study site during the same period as the lidar surveys. The imagery was collected using a DJI Phantom 4 RTK (Shenzhen, China) during leaf-off conditions. An orthomosaic of the study site was generated using Agisoft Metashape (version 1.8.4). Within the extent of forest cover in our study area, the Green Leaf Index (GLI, Equation (1); [34]) was calculated from the RGB imagery as

$$GLI = (2B_G - B_R - B_B)/(2B_G + B_R + B_B), \tag{1}$$

where $B_X$ is the digital number of the band, G, R, or B. During leaf-off conditions, GLI was used to distinguish conifer species from deciduous species.

To account for surrounding vegetation for any given grid cell and in situ sampling plots, the initial GLI output was coarsened to 10 m spatial resolution using average GLI per 10 m pixel. We used a threshold of 0.2 to define two classes within the forested area of our study site; the two classes were assumed to represent an approximate neighborhood of conifer vegetation (GLI ≥ 0.2) and leaf-off deciduous vegetation (GLI < 0.2). A field class was added by manual interpretation of the orthomosaic at the field/forest boundary. Figure 1 shows the RGB orthomosaic and the 10 m GLI map of the study site.

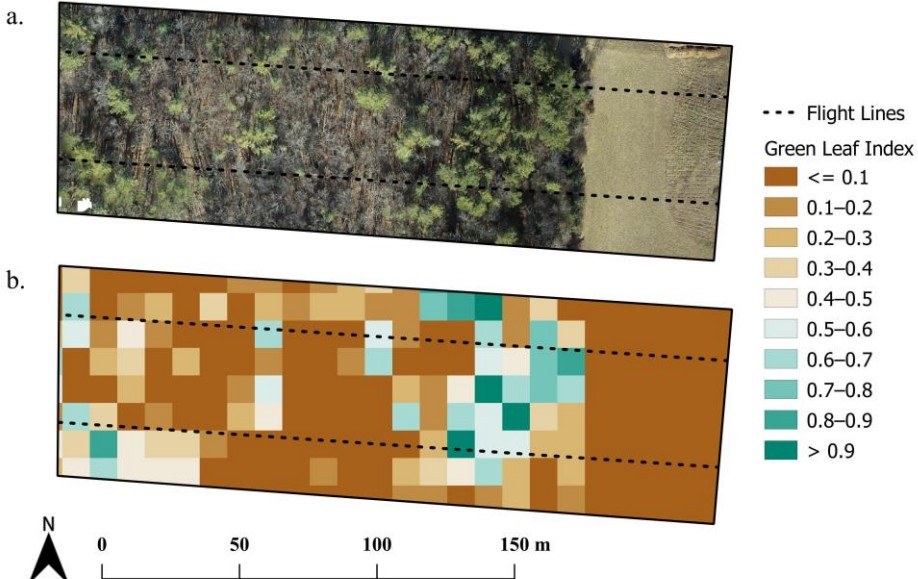

**Figure 1.** An optical orthomosaic of the study area (**a**) was used to produce a Green Leaf Index map (**b**) to derive vegetation classification. Any Green Leaf Index values < 0.2 were classified as deciduous canopy for our analysis, except for the manually classified field cells.

### 2.4. Snow Depth Estimates and Bias Evaluation

We measured snow depth in situ at 25 sampling locations along three parallel transects below the UAS flight path [35]. Measurements were completed immediately following UAS lidar surveys. The sampling locations were distributed across the three cover types defined by the GLI map: (1) open field (*n* = 3); (2) deciduous forest (*n* = 13); and (3) conifer forest (*n* = 9). At each sampling location, a square 1 m$^2$ quadrat was situated such that a corner of the grid fell on the transect with minimal disturbance of the snow surface. A corner of each grid cell was surveyed in using a Trimble© Geo7X Global Navigation Satellite System (GNSS) positioning unit with Zephyr™ antenna (Trimble, Westminster, CO, USA). Following differential correction using a nearby (~5 km) CORS (Continuously Operating Reference Station), an average estimated horizontal uncertainty of approximately 2, 5, 7 cm was observed for the field, deciduous, and coniferous sampling sites, respectively. Orientation and position of the quadrat was noted and photographed at each location for

retrieval of lidar returns within spatially explicit quadrat bounds. Within each quadrat, nine snow depth measurements were collected using a Federal snow tube sampler. Snow depth within the quadrat was calculated as the mean of its measurements, and 95% confidence intervals were also calculated.

From the lidar data, snow depth estimates were calculated for all combinations of flight speeds across the study site on a 1 m grid by subtracting the mean return elevation of the baseline survey from the mean return elevation of the snow-on survey. Likewise, at each sampling site, baseline and snow-on returns within the extent of each quadrat were used to calculate snow depth for comparison with in situ observations. Confidence intervals were calculated for each 1 m grid cell over the entire study site and within the quadrat located at each sampling site using the pooled standard deviation of the two samples, total return count accumulated between the two samples, and the pooled degrees of freedom [36]. We then characterized the distribution of confidence intervals by forest cover type and pairwise flight speed combinations across our study area. Next, based on reported snow depth errors in the field of 1 cm bias and 1.2 cm RMSD [26], we used a target precision of 2 cm to produce raster grids for each baseline flight speed, where the grid cell value reported the fastest snow-on flight speed that could be used to achieve the target snow depth estimate precision. This precision equates to a 10% error for snow depths of 20 cm. These maps were used to understand the effects of different flight speed combinations on snow depth retrieval and precision across different vegetation types.

For the comparison of lidar and in situ snow depths, we evaluated lidar snow depth retrievals for flight speed combinations using estimates of the aggregated snow depth mean difference by vegetation type. This error metric was selected because of the limited range of snow depth measurements for the sample date within and between vegetation types. Within forest grid cells, limited lidar returns may not be representative of any given grid cell and the return locations may differ between snow-on and snow-off flights. We characterized common coverage within grid cell lidar return distributions between the two datasets by calculating the total area within a plot that had ground retrievals from both the snow-on and snow-off surveys. This area of overlap was calculated in QGIS (v 3.26.1) by buffering each lidar return location by 5 cm (~the upper limit of the payload reported horizontal error), then calculating the area of the intersection of the buffers between the snow-off and snow-on data sets for each of the sixteen pairs of flight speeds.

## 3. Results

### 3.1. Lidar Snow Depth Estimate Precision

Snow depth precision was estimated using confidence intervals for all sixteen combinations of baseline and snow-on speed by land type. Across the study site, the distribution of snow depth confidence intervals for pairwise flight speed combinations had similar patterns for all land types (Figure 2). Flight speed had little impact on snow depth confidence intervals within the field, with nearly all grid cells producing snow depth confidence intervals less than 1.5 cm for all flight speed combinations. The narrowest confidence intervals were produced from flight speeds of 2 m s$^{-1}$ for both surveys for all three cover types (Table 1). In general, however, there were disparities between field and forest confidence intervals of snow depth, with consistently narrower confidence intervals in the field at any flight speed combination due to higher ground return density in the field compared to the forest.

Within deciduous and conifer grid cells, there was greater variability in snow depth confidence intervals among flight speed combinations (Figure 2), with reductions in the mean and median as flight speeds slowed for both baseline and snow-on flights. Like the disparities between field and forest confidence intervals, this effect is largely due to larger sample sizes at slower flight speeds. The narrowest confidence intervals occurred for the slowest flight speed combination (Table 1). For all flight speed combinations, there were narrower distributions of confidence intervals in deciduous areas with lower median values compared to conifer areas.

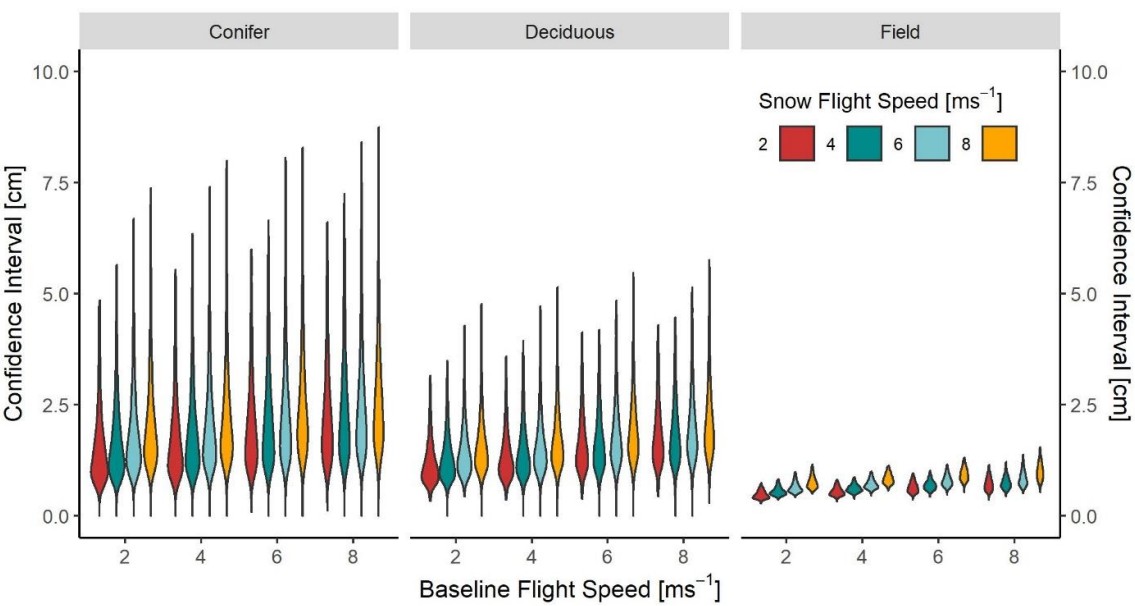

**Figure 2.** Violin plot of 150% of the interquantile range of confidence intervals for each cover type.

**Table 1.** Pairwise mean confidence interval (cm) by cover type.

| | | Baseline Flight Speed (m s$^{-1}$) | | | | |
|---|---|---|---|---|---|---|
| | | **2** | **4** | **6** | **8** | |
| Snow-on flight speed (m s$^{-1}$) | 2 | 2.87 | 3.42 | 3.96 | 4.32 | Conifer |
| | 4 | 3.45 | 3.99 | 4.56 | 4.92 | |
| | 6 | 3.74 | 4.18 | 5.21 | 5.22 | |
| | 8 | 4.04 | 4.58 | 5.26 | 5.42 | |
| | 2 | 2.04 | 2.20 | 2.66 | 2.96 | Deciduous |
| | 4 | 2.38 | 2.56 | 3.03 | 3.27 | |
| | 6 | 2.83 | 2.73 | 3.14 | 3.53 | |
| | 8 | 2.89 | 2.94 | 3.37 | 3.74 | |
| | 2 | 0.51 | 0.59 | 0.69 | 0.79 | Field |
| | 4 | 0.60 | 0.67 | 0.75 | 0.81 | |
| | 6 | 0.70 | 0.77 | 0.86 | 0.91 | |
| | 8 | 0.82 | 0.91 | 1.01 | 1.08 | |

*3.2. Flight Speeds to Achieve Precision Target*

Across the study area, reduced flight speeds for both baseline and snow-on flights resulted in lower mean confidence intervals. At a baseline flight speed of 2 m s$^{-1}$, to achieve a confidence interval of 2 cm or less, flight speeds of 8 m s$^{-1}$ could be used for a snow survey for >95% of the field region, >70% of the deciduous region, and <50% of the conifer region. At a faster baseline flight speed of 8 m s$^{-1}$ and snow-on flight speeds of 8 m s$^{-1}$, >95% of the field region still achieved 2 cm precision, while <45% of the deciduous region and <30% of the conifer region had a 2 cm confidence interval. Within open field areas, the highest flight speed combination tested resulted in <2 cm confidence intervals consistently, while deciduous and conifer areas resulted in <2 cm confidence intervals at progressively higher rates as flight speeds were reduced (Figure 3 and Table 2a).

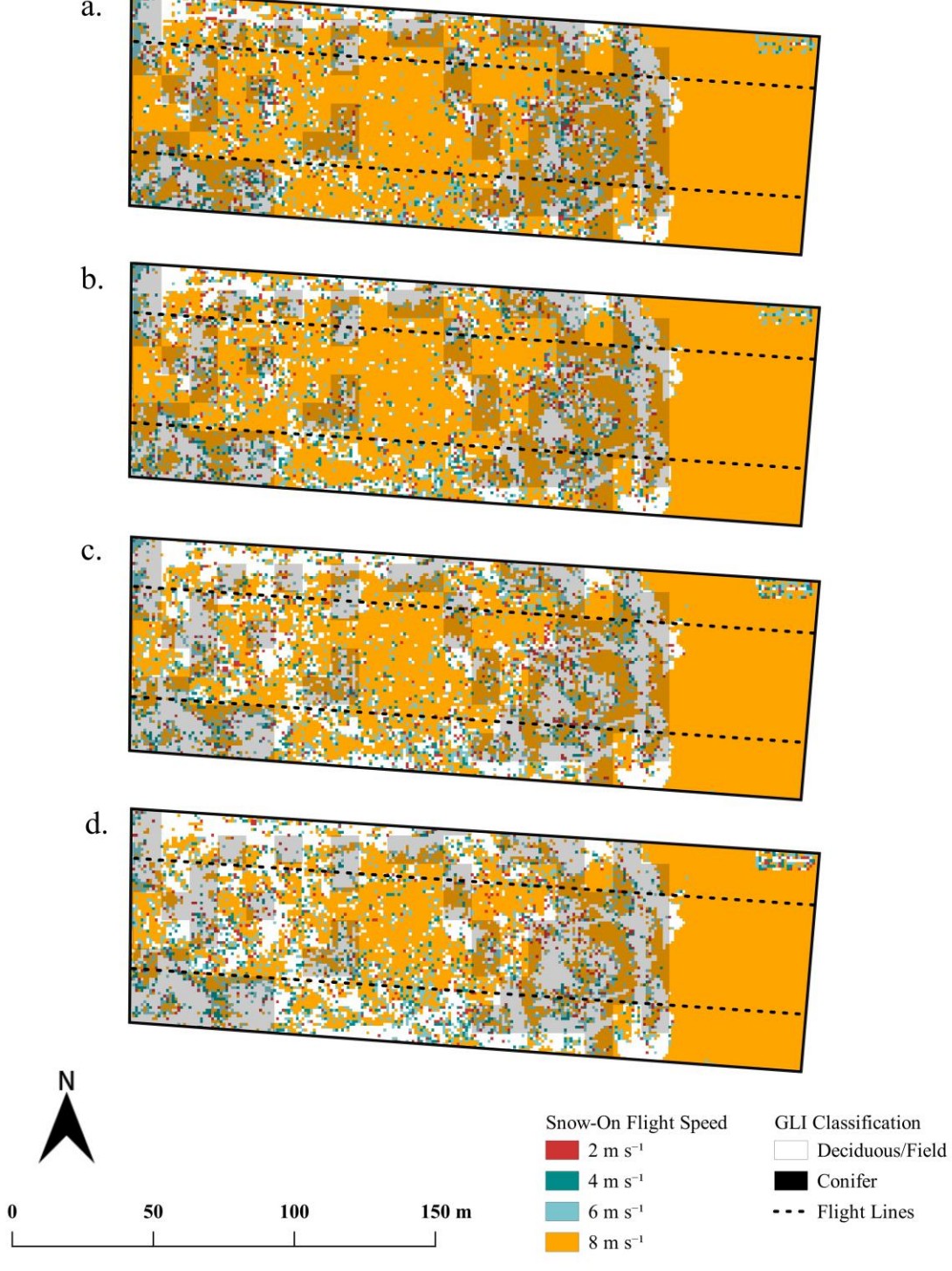

**Figure 3.** Maps of the study area colored by the maximum snow-on flight speed per pixel required to produce 2 cm estimate precision at baseline speeds of (**a**) 2 m s$^{-1}$, (**b**) 4 m s$^{-1}$, (**c**) 6 m s$^{-1}$, and (**d**) 8 m s$^{-1}$. The areas in white did not produce a 2 cm estimate precision at any flight speed.

Among grid cells in which we did not achieve <2 cm precision at any given flight speed combination, the majority were due to insufficient sample size needed to calculate confidence intervals (Table 2b), as opposed to an absence of data (Table 2c) or high variability. In the forest areas, Table 2b shows that the percentage of grid cells with insufficient sample sizes changed with both baseline flight speed and snow-on flight speed within a cover type, when either survey flight speed is held constant.

**Table 2.** Proportion of study area grid cells which have (**a**) achieved <2 cm precision, (**b**) produced estimate precision > 2 cm, and (**c**) have no snow depth estimate.

| | | | Baseline Flight Speed (m s$^{-1}$) | | | | |
|---|---|---|---|---|---|---|---|
| | | | **2** | **4** | **6** | **8** | |
| **a. <2 cm Precision** | Snow-on flight speed (m s$^{-1}$) | 2 | 0.706 | 0.645 | 0.536 | 0.470 | Conifer |
| | | 4 | 0.658 | 0.602 | 0.495 | 0.434 | |
| | | 6 | 0.600 | 0.541 | 0.442 | 0.381 | |
| | | 8 | 0.494 | 0.435 | 0.334 | 0.282 | |
| | | 2 | 0.851 | 0.802 | 0.709 | 0.631 | Deciduous |
| | | 4 | 0.820 | 0.771 | 0.675 | 0.597 | |
| | | 6 | 0.780 | 0.727 | 0.623 | 0.538 | |
| | | 8 | 0.708 | 0.644 | 0.531 | 0.443 | |
| | | 2 | 0.987 | 0.984 | 0.982 | 0.975 | Field |
| | | 4 | 0.985 | 0.984 | 0.977 | 0.969 | |
| | | 6 | 0.982 | 0.980 | 0.971 | 0.962 | |
| | | 8 | 0.970 | 0.969 | 0.961 | 0.954 | |
| **b. >2 cm Precision** | | 2 | 0.272 | 0.331 | 0.430 | 0.488 | Conifer |
| | | 4 | 0.320 | 0.374 | 0.470 | 0.525 | |
| | | 6 | 0.377 | 0.432 | 0.522 | 0.575 | |
| | | 8 | 0.479 | 0.535 | 0.626 | 0.672 | |
| | | 2 | 0.141 | 0.188 | 0.277 | 0.350 | Deciduous |
| | | 4 | 0.172 | 0.218 | 0.311 | 0.383 | |
| | | 6 | 0.211 | 0.262 | 0.361 | 0.441 | |
| | | 8 | 0.284 | 0.346 | 0.455 | 0.538 | |
| | | 2 | 0.013 | 0.015 | 0.018 | 0.025 | Field |
| | | 4 | 0.014 | 0.016 | 0.022 | 0.030 | |
| | | 6 | 0.017 | 0.020 | 0.028 | 0.037 | |
| | | 8 | 0.029 | 0.030 | 0.038 | 0.045 | |
| **c. No Snow Depth Estimate** | | 2 | 0.022 | 0.024 | 0.035 | 0.042 | Conifer |
| | | 4 | 0.022 | 0.024 | 0.034 | 0.041 | |
| | | 6 | 0.023 | 0.026 | 0.036 | 0.044 | |
| | | 8 | 0.026 | 0.030 | 0.039 | 0.046 | |
| | | 2 | 0.008 | 0.010 | 0.014 | 0.020 | Deciduous |
| | | 4 | 0.008 | 0.011 | 0.014 | 0.020 | |
| | | 6 | 0.009 | 0.011 | 0.015 | 0.020 | |
| | | 8 | 0.008 | 0.010 | 0.014 | 0.020 | |
| | | 2 | 0.001 | 0.000 | 0.000 | 0.001 | Field |
| | | 4 | 0.001 | 0.001 | 0.001 | 0.001 | |
| | | 6 | 0.001 | 0.001 | 0.001 | 0.001 | |
| | | 8 | 0.001 | 0.001 | 0.001 | 0.001 | |

While achieving high precision snow depth information is important, grid cells with no lidar returns for either the baseline or snow-off flight result in a gap in the mapped snow depth that remains as a no-data value or must be gap-filled. At high flight speeds,

nearly 5% of the coniferous forest is not mapped. The relative change in the percent of grid cells was not consistent with survey flight speed changes. For both forest cover types, the snow-on survey speed increase from 6 m s$^{-1}$ to 8 m s$^{-1}$ had the greatest reduction in target precision coverage due to insufficient sample size, with an average change in coverage of 10.2% for all baseline flight speeds in conifer areas and 8.7% for all baseline flight speeds in deciduous areas. For baseline surveys, the increase in flight speed from 4 m s$^{-1}$ to 6 m s$^{-1}$ was associated with the greatest reduction in target precision coverage due to insufficient sample size, with an average change in coverage of 9.4% for all baseline flight speeds in coniferous areas and 9.7% for all baseline flight speeds in deciduous areas. On the other hand, Table 2c shows that there is little change in the percentage of grid cells without a snow depth estimate when baseline flight speed is held constant. In other words, if no snow depth estimate could be made for a grid cell, it was generally due to the absence of baseline ground returns rather than snow-on ground returns. We hypothesize that this is a consequence of the different reflectance and surface scattering attributes of snow and leaf litter and the vegetation canopy leaf area that resulted in an increased point density over the snow surface as compared to the bare-earth surface (Table A1). Within the field, the lidar return densities were similar for both snow-on and snow-free surveys. This observation demonstrates the increase in absorption and scattering caused by forest debris (Table A2). For conifer areas, the increase in baseline flight speed from 4 m s$^{-1}$ to 6 m s$^{-1}$ was associated with the greatest increase in no snow depth estimate grid cells with an average increase of 1%, while in deciduous areas, the increase in flight speed from 6 m s$^{-1}$ to 8 m s$^{-1}$ caused the greatest increase in no snow depth estimate grid cells with an average increase of 0.5%.

### 3.3. Lidar vs. In Situ Snow Depth Estimation

Across the study site, in situ snow depth measurements ranged from 9.9 to 15.4 cm for the three open field, 13 deciduous forest, and nine conifer forest sampling locations. In general, we observed good agreement between in situ and lidar-derived snow depths across our study site, with outliers most prevalent and poorer agreement at baseline flight speeds of 2 m s$^{-1}$ and 8 m s$^{-1}$. The mean difference and RMSD do not exhibit organized changes by baseline and snow-on flight velocity. Within vegetation cover types, we generally observed a higher mean difference in conifer areas compared to deciduous areas, with a broader interquartile range (Figure 4). The conifer areas' interquartile ranges are tighter for lower baseline speeds, but outliers are still evident. However, for most flight speed combinations and vegetation cover types, the mean difference was not significantly different from zero (Table 3), but the 2 m s$^{-1}$ baseline speeds resulted in differences that were on the order of one cm higher than differences for the 8 m s$^{-1}$ speeds.

When lidar returns are sparse and only a portion of the grid is measured, it is possible that the baseline and snow-on UAS lidar maps capture different areas within a grid. If there are elevation changes or local surface anomalies (e.g., downed trees or rocks), then these differences could cause considerable snow depth errors. The baseline and snow-on lidar return locations were investigated at in situ sampling locations to understand the extent to which the returns measured elevations at the same location within a single plot (e.g., Figure 5). At the plot scale within forest cover types, reducing the flight speed dramatically increased the overlap of baseline and snow-on point clouds (Figure 6). At the slowest speeds, the overlap rarely exceeded 60%, and the fastest flights had less than a 20% overlap. Regardless, there is still a considerable portion of grid cells where the baseline and snow-on sampling locations are not coincident for all flight speeds. Figure 6 also shows that reductions to the baseline flight speed have a greater impact on overlap than equivalent speed changes to the snow-on flights.

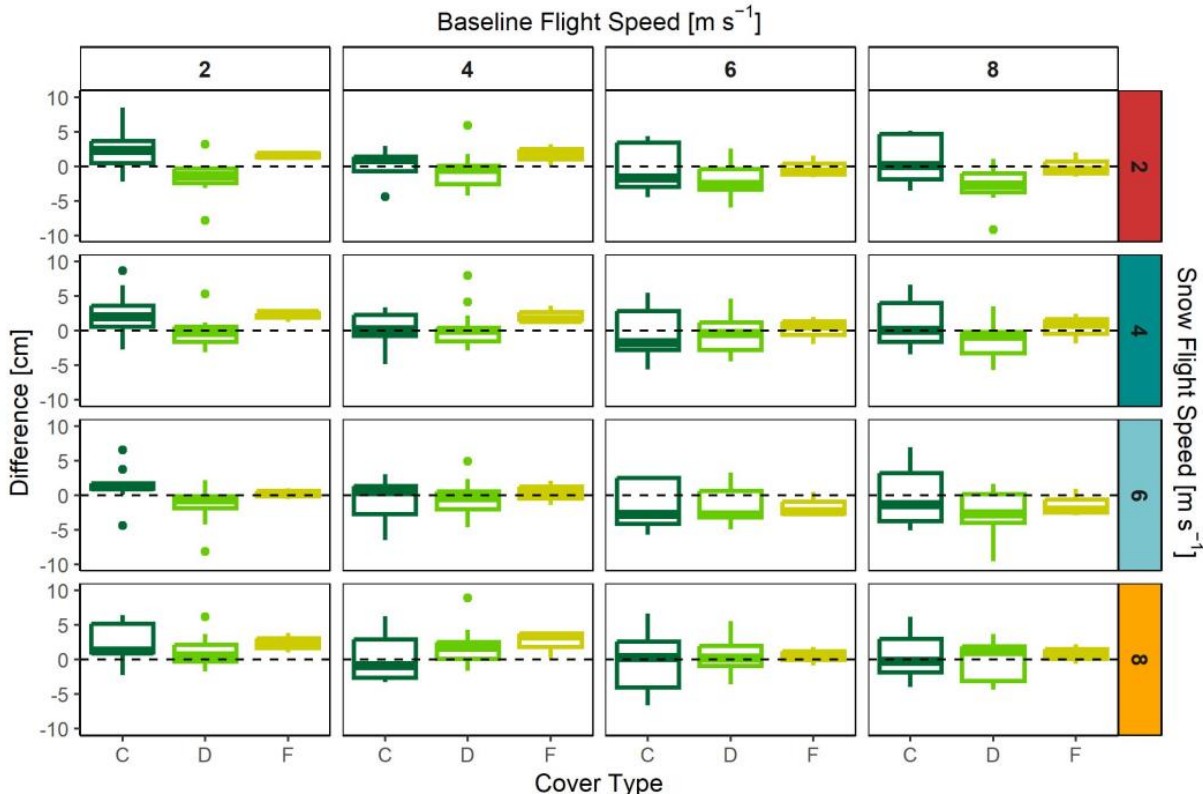

**Figure 4.** Box plots of the differences between lidar-derived snow depth and in situ snow depth for all flight speed combinations by field (F), deciduous (D), and coniferous (C) vegetation cover types. The box is framed by lower and upper quartile values with its center line indicating the median value. The box height is the interquartile range (IQR). The lower whisker extends to 1.5 times the IQR below the lower quartile or the minimum value in the dataset. The upper whisker extends to 1.5 times the IQR above the upper quartile or the maximum value in the dataset. Outliers are also shown.

**Table 3.** Mean difference (**a**) and root mean square difference (**b**) for cover types by flight speed, with means significantly different from zero in bold italics.

| | | Baseline Flight Speed (m s$^{-1}$) | | | | | | | | |
|---|---|---|---|---|---|---|---|---|---|---|
| | | a. Mean Difference | | | | b. RMSD | | | | |
| | | 2 | 4 | 6 | 8 | 2 | 4 | 6 | 8 | |
| Snow-on flight speed (m s$^{-1}$) | 2 | ***2.43*** | 0.33 | −0.16 | 0.78 | 3.78 | 2.12 | 3.33 | 3.33 | Conifer |
| | 4 | 2.35 | 0.26 | −0.23 | 0.71 | 4.12 | 2.36 | 3.66 | 3.46 | |
| | 6 | 1.32 | −0.77 | −1.27 | −0.32 | 3.05 | 3.01 | 3.48 | 3.87 | |
| | 8 | 2.37 | 0.28 | −0.21 | 0.73 | 3.78 | 3.33 | 4.27 | 3.14 | |
| | 2 | −1.48 | −0.78 | ***−1.97*** | ***−2.75*** | 2.79 | 2.77 | 3.03 | 3.65 | Deciduous |
| | 4 | −0.22 | 0.48 | −0.71 | ***−1.48*** | 2.00 | 2.91 | 2.61 | 2.77 | |
| | 6 | −1.28 | −0.59 | ***−1.78*** | ***−2.55*** | 2.89 | 2.52 | 3.07 | 4.09 | |
| | 8 | 1.10 | ***1.79*** | 0.60 | −0.17 | 2.37 | 3.14 | 2.74 | 2.63 | |
| | 2 | ***1.58*** | 1.72 | −0.29 | −0.04 | 1.64 | 2.12 | 1.37 | 1.50 | Field |
| | 4 | ***2.12*** | 2.26 | 0.25 | 0.50 | 2.20 | 2.46 | 1.63 | 1.84 | |
| | 6 | 0.20 | 0.34 | −1.67 | −1.42 | 0.65 | 1.47 | 2.23 | 2.17 | |
| | 8 | 2.39 | 2.53 | 0.52 | 0.77 | 2.65 | 3.05 | 1.18 | 1.39 | |

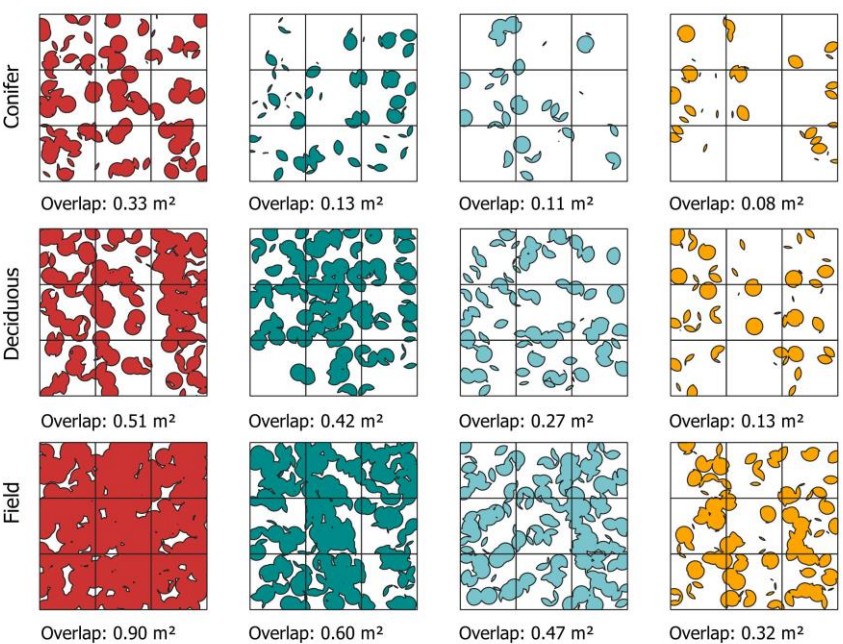

**Figure 5.** Examples of areas of overlap metrics between baseline and snow-off flights by land use. Each $1 \times 1$ m grid cell was divided into nine subplots. The total area of overlap was calculated for the entire grid cell. For consistency, colors used here are for snow-on flight speeds of $2$ m s$^{-1}$ (red), $4$ m s$^{-1}$ (blue), $6$ m s$^{-1}$ (light blue), and $8$ m s$^{-1}$ (gold).

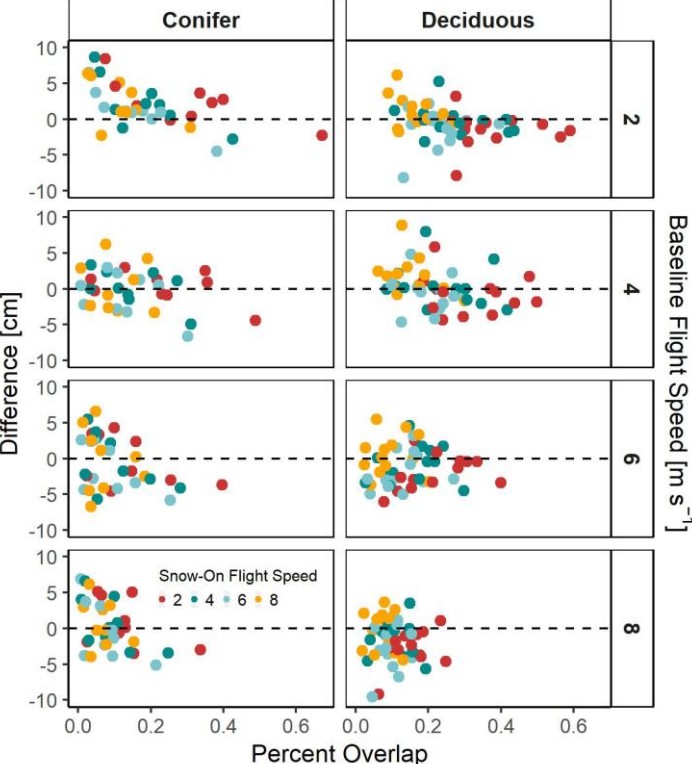

**Figure 6.** Snow depth difference between in situ and lidar observations vs. percent overlap for conifer and deciduous study plots for all baseline flight speeds, colored by snow-on flight speeds of $2$ m s$^{-1}$ (red), $4$ m s$^{-1}$ (blue), $6$ m s$^{-1}$ (light blue), and $8$ m s$^{-1}$ (gold). In each figure, there are four points for each individual grid cell corresponding to the overlap and bias determined for each of the four snow-on flight speeds.

Although the overlap increased with reduced flight speeds for either survey, the minimum mean difference did not correspond with the greatest mean overlap for either cover type (refer to Figure 4). Instead, the minimum absolute mean difference corresponded with a baseline flight speed of 6 m s$^{-1}$ and a snow-on flight speed of 2 m s$^{-1}$ for conifer plots and a baseline flight speed of 8 m s$^{-1}$ and a snow-on flight speed of 8 m s$^{-1}$ for deciduous plots. However, the range of differences tightens with an increase in overlap.

## 4. Discussion

For flight planning over a mixed landscape, the ideal flight speed combination used in snow depth estimate surveys is one that results in a point density that can provide a representative sample of the surface conditions with an acceptable number of low-data availability pixels [23]. There is clear evidence that reducing speeds increases precision and decreases low data pixels. It is also clear that the baseline flight speed is more critical than the snow-on flight speed.

Both the baseline survey and the snow-on survey influence snow depth estimate bias, which, when compared to observations, should both minimize the RMSD and have a mean difference that is close to and not significantly different from zero. If the goal is to minimize the mean difference, the best flight combinations would differ. In this case, a baseline flight of 6 m s$^{-1}$ combined with a snow-on flight of 2 m s$^{-1}$ (−0.16 cm), a baseline flight of 8 m s$^{-1}$ combined with a snow-on flight of 8 m s$^{-1}$ (−0.17 cm), and a baseline flight of 8 m s$^{-1}$ combined with a snow-on of 2 m s$^{-1}$ (−0.04 cm) for coniferous, deciduous, and fields, respectively. However, this might not be advisable in study areas where there is potential for overprobing by the snow tubes [35]. Because the RMSD metric amplifies the differences among outliers, overall consistency may be achieved when the RMSD is minimized. From the minimum RMSD, the best flight combinations are: baseline of 4 m s$^{-1}$ and snow-on 2 m s$^{-1}$ (2.12 cm); baseline of 2 m s$^{-1}$ and snow-on 4 m s$^{-1}$ (2.00 cm); baseline of 2 m s$^{-1}$ and snow-on of 6 m s$^{-1}$ (0.65 cm) for coniferous, deciduous, and fields, respectively. These RMSD-derived baseline flight speeds are better aligned with those with reduced precision.

From this information, the results presented here suggest that there are several variable flight speed combinations that would promote appropriate difference distributions across vegetation types, with similar RMSD and absolute mean differences of the same order of magnitude. However, the mean rank of these two measures in combination provides a characterization of these metrics together. By mean rank, flight speeds of 4 m s$^{-1}$ for both surveys provided the best balance of mean differences and their distribution over conifer vegetation. On the contrary, over deciduous vegetation, a slower baseline flight speed of 2 m s$^{-1}$ with a snow-on flight speed of 4 m s$^{-1}$ had the lowest mean rank. Although the field cover type had a low sample size ($n = 3$) in this study, by mean rank, the results were consistent with deciduous vegetation cover in that a slower baseline flight speed of 2 m s$^{-1}$ combined with a faster snow-on flight speed of 6 m s$^{-1}$ had the lowest mean rank and produced appropriate differences in distributions and absolute mean differences. With these flight speed combinations, we found a mean difference of 0.007 cm with a 95% confidence interval range of 1.712 cm across all vegetation cover types. With an intercept forced to zero, lidar and in situ snow depth estimates were strongly correlated with a slope close to 1 and lower AIC than a free intercept model (RMSE = 2.03 cm, Figure 7).

Intuitively, acquiring the baseline ground survey at a slower flight speed makes sense. Because of differences in the light scattering characteristics of snow and dead vegetation, surface reflectance in the near infrared at the wavelength of the sensor used for this study (905 nm) is relatively low for vegetation litter compared to a smoother snow surface, which likely influenced an increased number of ground returns for snow-on conditions. In contrast, Feng et al.'s finding of lower ground returns for snow-on conditions may be due to surface conditions that result in lower reflectivity, as reflectance is reduced when the snow surface is wet or old. This represents an important consideration for UAS operators evaluating snow depth, especially for the purpose of snowpack evolution. As snow ages, grain size increases [37], and impurities are introduced to the snow surface [38], both

of which decrease reflectance at 905 nm [39–41]. For New Hampshire snowpacks, grain size largely controls albedo [42]. At warmer temperatures, wetter snowpacks can also become problematic for lidar measurements because of strong water absorption in the NIR [43,44], especially at 1550 nm compared to 905 nm, two common wavelengths for lidar sensors. We hypothesize that any reduction in NIR reflectance caused by aging snow or ambient air temperatures allowing for surface melt should require slower flight speeds to accommodate ground retrieval data needs for accurate snow depth maps. Additionally, laser pulses are lost at a higher rate in forest canopies than open ground [26,32], which provides one possible explanation for the finding in this study that snow surveys should be conducted at a higher flight speed in open fields compared to forests, where we would expect to have a lower ground return density in both surveys at any stable flight speed.

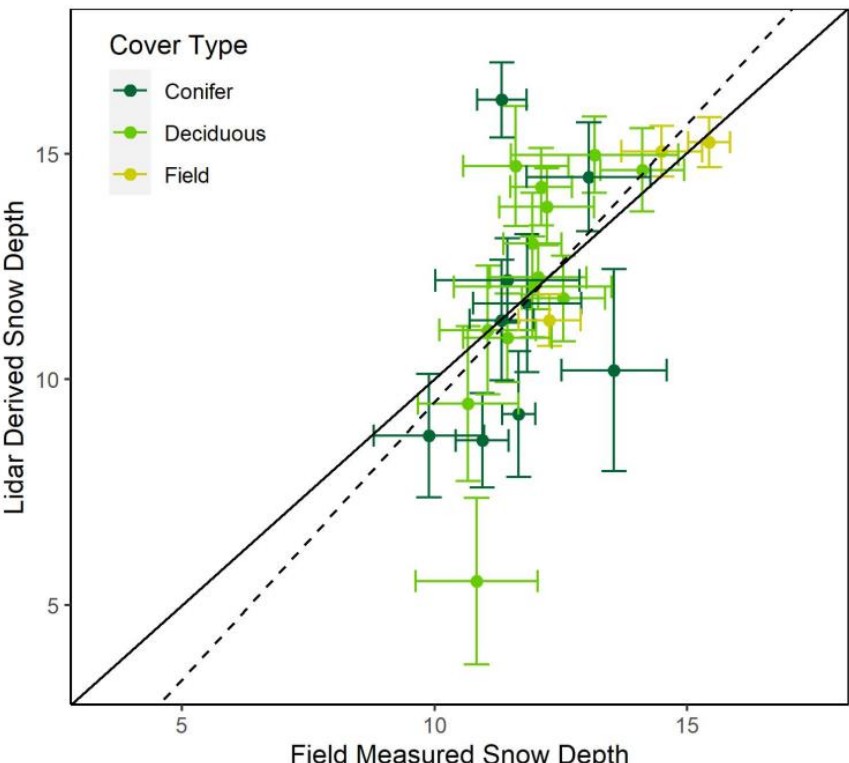

**Figure 7.** Scatter plot of lidar-derived and in situ snow depth estimates with 95% confidence intervals shown for each measurement, using flight speed combinations that minimized the mean differences between methods, with a model forced to zero intercept (solid black line) and a model with a free intercept (dashed line).

Our result in conifer types was less than expected. Mean absolute differences in snow depth estimates may have resulted from imprecise in situ measurements or from lidar estimates. In situ snow depth measurements in forest environments are subject to overestimation caused by the measurement tool penetrating into the leaf litter and soil surface [35]. In this study, we sought to limit this issue by retrieving snow depth measurements with a Federal snow tube as opposed to a Magnaprobe. Nevertheless, in situ snow measurements may have been imprecise or overestimated in some cases in our study, especially for forest plots, which would have contributed to errors. From the standpoint of lidar measurements, we hypothesize that the result is due to the oversampling of non-overlapping areas that occurred during a slow 2 m s$^{-1}$ baseline survey in conifer stands. Oversampling, combined with the presence of subtle terrain variation, could lead to an over- or under-estimate of the baseline mean elevation, which would carry through to snow depth estimation. The relative change from baseline flight speed of 4 m s$^{-1}$ to 2 m s$^{-1}$ in Table 3a provides some evidence in support of this concept. Here, a reduction of the

baseline flight speed was associated with a large increase in the absolute mean difference, and specifically an increase in the difference by 2–2.5 cm, suggesting that on average, a 2 m s$^{-1}$ baseline flight speed results in a lower mean baseline elevation compared to a 4 m s$^{-1}$ baseline flight speed in coniferous areas. In this study, we evaluated sample overlap and distribution between all flight speeds at each survey, and the results were inconclusive. We did not evaluate non-overlapping samples or assess the effect they had on mean elevation estimates.

In summary, the value of the mean difference is diminished, and for a baseline survey, in areas with minimal canopy foliage (e.g., bare deciduous or open fields), we recommend a flight speed of 2 m s$^{-1}$ and in areas with more canopy foliage, such as a conifer area, we recommend a flight speed of 4 m s$^{-1}$. During a snow-on survey, the best distribution of snow depth estimates is then achieved with flight speeds of 6 m s$^{-1}$ over open fields and between 2 and 4 m s$^{-1}$ over forest areas.

## 5. Conclusions

We found that in a mixed landscape, the expected mean and variation in measurement differences can be reduced by adjusting survey flight speed depending on vegetation type and ground surface conditions. The outcome of our analysis provided a set of flight speeds that are convenient in that there are only two different speeds that we recommend during each survey. Modifying flight speeds to account for patterns of bias in snow depth estimation had not previously been explored. Our results demonstrate that variable flight speeds within and between UAS lidar surveys based on vegetation cover and ground conditions can be used to balance efficiency and data quality. Although this study was conducted in a mixed forest landscape, the results may be applicable to other forested regions with shorter stature trees (<30 m). Taller vegetation or higher-density forest types, especially those with longer leaf longevity, may represent additional challenges for ground surface detection due to sensor range limitations and signal attenuation in forest canopies. Generally, consumer-grade UAS-borne lidar sensors have similar capabilities in terms of detection range but differ in sampling pattern. The sensor used for this study, the VLP-16, has a moderately higher effective field-of-view compared to other lidar sensors, like the LiVox Avia. Because of this, rates of ground detection may differ, especially under dense needleleaf canopies where a high observation angle could be advantageous. During flight planning, UAS operators should consider environmental conditions, especially vegetation composition and snow characteristics, to determine optimal lidar sampling strategies for snow depth estimation.

**Author Contributions:** Conceptualization, F.B.S., A.G.H. and J.M.J.; methodology, F.B.S., A.G.H. and M.W.P.; formal analysis, F.B.S. and A.G.H.; investigation, F.B.S. and A.G.H.; data curation, F.B.S. and A.G.H.; writing—original draft preparation, F.B.S. and A.G.H.; writing—review and editing, F.B.S., A.G.H., J.M.J. and M.W.P.; software, F.B.S. and A.G.H.; visualization, F.B.S.; funding acquisition, M.W.P. and J.M.J. All authors have read and agreed to the published version of the manuscript.

**Funding:** Approved for Public Release. Distribution is Unlimited. This material is based upon work supported by the Broad Agency Announcement Program and the Cold Regions Research and Engineering Laboratory (ERDC-CRREL) under Contract No. W913E518C0005 and W913E521C0006.

**Data Availability Statement:** Data used to conduct this experiment will be made available upon request to the authors until it is published.

**Conflicts of Interest:** The authors declare no conflict of interest. Any opinions, findings and conclusions or recommendations expressed in this material are those of the author(s) and do not necessarily reflect the views of the Broad Agency Announcement Program and ERDC-CRREL.

## Appendix A

**Table A1.** Average number of lidar ground returns per 1 m$^2$ by land type, flight speed, and snow conditions.

| | Baseline | | | | Snow-on | | | |
|---|---|---|---|---|---|---|---|---|
| **Flight Speed (m s$^{-1}$)** | **2** | **4** | **6** | **8** | **2** | **4** | **6** | **8** |
| Conifer | 129 | 88 | 55 | 41 | 350 | 178 | 117 | 88 |
| Deciduous | 197 | 133 | 85 | 63 | 542 | 272 | 184 | 138 |
| Field | 918 | 611 | 390 | 273 | 1194 | 592 | 402 | 304 |

**Table A2.** Percent increase in the average number of lidar ground returns per 1 m$^2$ for snow-on compared to bare-earth conditions by land type, flight speed, and snow conditions.

| | Percent Increase in Ground Returns | | | |
|---|---|---|---|---|
| **Flight Speed (m s$^{-1}$)** | **2** | **4** | **6** | **8** |
| Conifer | 170% | 103% | 112% | 113% |
| Deciduous | 176% | 105% | 117% | 120% |
| Field | 30% | −3% | 3% | 11% |

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
