# Peer review of "Evaluating the Effects of UAS Flight Speed on Lidar Snow Depth Estimation in a Heterogeneous Landscape"

_remotesensing, doi:10.3390/rs15215091_

Round 1
Reviewer 1 Report
Study the snow depth using lidars, how do the flight speed affect.
Very thoroughly presented work, large amount of data (could ask if even less would be sufficient, but now all is done).
Idea seems to be to measure the baseline surface in snow free conditions, and the snow surface in snow condition, and subtract to get the snow depth. This is logical arrangement, though one could ask, how well the lidar position is known, assuming conditions are different, and how stable the baseline is on soft surface, e.g. swamp, where the changes in ground water level, freezing of the soil, and pressure of the snow can change the surface sometimes quite a lot.
One could also ask, how realistic the 2 cm precision is in practise, if both the baseline surface and snow layer surface undulates in much larger scales. Or how uniquely the baseline can even be measured under vegetation.
And if there is a dependence on snow amount and surface patch visibility, e..g. by snow accumulation and melting.
Maybe also make a fast search on vegetation and snow spectra, their dependence on season, wetness, snow grain size for the discussion part.
The conclusions leaves a bit so-what feeling. How can other uav operators use the results? What if different instrumenst? Other region? Other forest type?
I can happily recommend publications, with optional revision on points above.
Reviewer 2 Report
This works intends to provide a guide to choose the appropriate flight speed in UAV Lidar surveys to determine snow depth. The results should be of interest to all the researchers using this technology.
The methodology is well explained and the results are clearly understood.
However, the conclusions the authors draw from the results are not clear.
I do not see how they can choose any appropriate flight speed from the results.
I think the main conclusions can be extracted form Table 3. As the authors state, the appropriate flight velocity must minimize the mean difference and the RMSD of lidar vs in-situ snow depth.
From Table 3, it seems that:
1.- Both the mean difference and RMSD vary randomly with both basal and snow-on flights velocity.
2- It seems that the conclusions that can be drawn from minimum mean difference differ from those obtained from minimum RMSD.
From the minimum mean difference, the best flights combinations are: baseline flight of 8 m/s combined with a snow-on flight of 6 m/s (-0.32); baseline flight of 8 m/s combined with a snow-on flight of 8 m/s (-0.17); baseline flight of 8 m/s combined with a snow-on of 2 m/s (-0.04).
From the minimum RMSD, the best flights combinations are: baseline of 4 m/s and snow-on 2 m/s (2.12); baseline 2 m/s and snow-on 4 m/s (2.00); baseline of 2 m/s and snow-on of 6 m/s (0.65).
3.- According to the results shown in point 2 above, all the conclusions of the authors are wrong.
The authors must work on the conclusions, since it is not clear to see the conclusions they state on the paper.
Minor comment: in the caption of Figure 4, the authors refer to the class forest. Is it field, instead?
Reviewer 3 Report
The reviewer would like to thank the author for this thoughtful manuscript. This work has good potential. The author is requested to put in some additional efforts to improve the quality of this manuscript.
Introduction
With the concern of the ongoing climate change related environmental changes, the authors must highlight the multiple use of their proposed technique for monitoring the variations in the high mountain cryosphere. Please discuss the scope of this work and cite the following article to highlight the significance in terms of monitoring natural disaster related changes with the proposed technique.
-Shugar et al, A massive rock and ice avalanche caused the 2021 disaster at Chamoli, Indian Himalaya, Science, 2021.
Snow Albedo in Forest
The authors are requested to discuss the issue of handling snow albedo in boreal forests and cite the following articles in the discussion. Please highlight the correlation of the results presented in the following articles with the findings of these investigations. Indicate the findings of the shadowed areas being compensated by higher solar illumination for better snow cover detection.
-Kostadinov et al., 2019. “Watershed-scale mapping of fractional snow cover under conifer forest canopy using lidar”, RSE.
-Muhuri et al., 2021. Performance Assessment of Optical Satellite-Based Operational Snow Cover Monitoring Algorithms in Forested Landscapes, IEEE JSTARS.
It is important to highlight here that the study indicated that lidar derived TCD does not bring any extra benefit over the satellite derived TCD.
Satellite vs. Ground/Airborne Derived TCD
The authors are requested to present a discussion on satellite vs. ground derived tree cover density (TCD).
Impact of Spatial Resolution
The authors are requested to comment on the impact of spatial resolution on this study.
Conclusion
The author is requested to list the key contributions in this section. At the moment the section is not detailed enough.
A grammatical check is desirable before the final publication.
Author Response
Please see the attachment. We are grateful that the reviewer took the time to review our manuscript, but we feel that many of these suggestions are minimally relevant or entirely outside of the scope of our work.

Round 2
Reviewer 2 Report
The authors have improved the manuscript following the reviewer's comments.